# Opioid use as a potential risk factor for pancreatic cancer in the United States: An analysis of state and national level databases

Usman Barlass[1], Ameya Deshmukh[2], Todd Beck[3], Faraz Bishehsari[1]*

**1** Division of Gastroenterology, Department of Internal Medicine, Rush University Medical Center, Chicago, IL, United States of America, **2** Midwestern University–CCOM, Downers Grove, IL, United States of America, **3** Bioinformatics and Biostatistics Core, Rush University Medical Center, Chicago, IL, United States of America

☯ These authors contributed equally to this work.
* faraz_bishehsari@rush.edu

**Data Availability Statement:** All relevant data are within the paper and its Supporting Information files.

## Abstract

Pancreatic cancer (PC) rate is increasing in the U.S. The use of prescription and illicit opioids has continued to rise nationally in recent years as well. Opioids have been shown to have a deleterious effect on multiple types of cancer with recent data suggesting opium use as a risk factor for PC. Using national databases, we tested whether opioid usage pattern over time could explain the state and national-based variations in PC rates in the U.S. Opioid death rate (as a surrogate for prescription and illicit opioid use) was extracted from the CDCs Wonder online data through the Vital Statistics Cooperative Program. Incidence of pancreatic cancer was retrieved from the online CDCs data base gathered from the U.S. Cancer Statistics Working Group. Prevalence of obesity, tobacco and alcohol use was collected from Behavioral risk factor surveillance system. Mixed-effects regression models were used to test the association between levels of PC rate and opioid death/use rates during the years 1999–2016. A rise in PC was seen over time at the national and state levels. Similarly, the opioid death rates increased over time. Among other potential PC risk factors, only obesity prevalence showed an increase during the study period. A state's opioid death rate at 4 years prior significantly predicted initial incidence of PC (β = 0.1848, p<0.0001) and had a significant effect on the estimated annual change in the rate of PC (β = -.0193, p<0.0001). Opioid use may be an un-identified risk factor contributing to the increasing incidence of PC in the U.S. These novel findings need to be verified by population-based studies.

## Introduction

Pancreatic cancer (PC) is currently the third leading cause of cancer related mortality in the United States (U.S) [1]. The five year survival rate for PC remains dismal (~10%), partly due to asymptomatic progression of the disease with a majority (~85%) of the patients having non-surgically resectable disease at the time of diagnosis [2]. More alarming is the ongoing increase

**Funding:** F.B. is supported by grants from the Rush Translational Sciences Consortium/Swim Across America Organization, Brinson Foundation, National Institutes of Health grant AA025387 and Institute for Translational Medicine (ITM), the National Center for Advancing Translational Sciences of the National Institutes of Health (NIH) (grant No. 5UL1TR002389–02) that funds the ITM. The content is solely the responsibility of the authors and does not necessarily represent the official views of the NIH. The funders had no role in study design, data collection and analysis, decision to publish, or preparation of the manuscript.

**Competing interests:** The authors have declared that no competing interests exist.

**Abbreviations:** PC, Pancreatic Cancer; U.S, United States; CDC, Centers for Disease Control and Prevention.

in the PC incidence which will place the disease as the second cause of cancer mortality by the end of this decade [3]. Discovering risk factors contributing to PC can aid with identifying high risk groups that would benefit from implementation of preventive and possibly screening strategies, which could help in reduction of the disease burden and/or early diagnosis.

While the established risk factors for PC including genetic predisposition syndromes and chronic pancreatitis (alcohol or non-alcohol causes) only account for a small portion of PC incident cases, the etiology for the majority of cases has remained unknown. Several factors associated with our lifestyle habits (e.g., smoking) have been proposed to modulate the risk of PC [1, 4–6]. Besides increasing obesity rates, exposure to a number of PC risk factors, such as smoking and alcohol, has been steady or even decreasing at the population level, contrary to the increase in the PC rates in the recent years. Therefore, novel factors may be at play.

Recent population based studies have suggested opium use to increase risk of PC in a dose dependent manner [7, 8]. While opium use is not a common recreational habit in the U.S., opioid use has been rising remarkably over the past decade. In fact, opioid misuse and over-dose have evolved into a public health crisis. Approximately 70,000 drug overdose deaths were reported in 2017, 68% of which involved an opioid [9]. The use of prescription opioids for the management of chronic pain has increased remarkably with more than 191 million opioid pre-scriptions given to patients in the U.S in 2017. Not surprisingly, the opioid addiction rates among patients who are given opioid for chronic pain has increased with 29% of such patients misusing opioids, and 12% developing an opioid use disorder [10].

Using national datasets, we aimed to evaluate whether the trend in the burden of opioid usage could explain the trend in PC at the national and state levels overtime.

## Methods

### Data collection

Pancreatic cancer incidence rates (age adjusted to the 2000 US standard population) from 1999 to 2016 for all 50 states and the District of Columbia and nationally were obtained from the CDC's U.S Cancer Statistics database (accessed November, 2020) [11]. These data were retrieved from the CDC's publicly available national database through their behavioral risk fac-tor surveillance system (BRFSS) (S1–S5 Tables). Upper and lower 95% confidence intervals (CIs) were retrieved for each risk factor from the existing datasets.

Opioid death rate data originated from the CDC wonder online database (released 2018) and retrieved through the Kaiser Family Foundation analysis (accessed November, 2020) [12]. Opioid death rate was used as a surrogate marker for overall opioid use to account for both prescription and illicit opioid use. Data are from the Multiple Cause of Death Files, 1999–2018, through the Vital Statistics Cooperative Program, as compiled from data provided by the 57 vital statistics jurisdictions. Drug overdose deaths were classified using the International Classification of Disease, Tenth Revision (ICD-10). All data was fully anonymized before it was accessed.

In order to correct for other possible proposed risk factors for PC, current cigarette smok-ing prevalence (1999 to 2017), obesity prevalence (1999 to 2018) and alcohol use prevalence (1999 to 2018) were also collected [13]. Obesity was defined as body mass index (BMI) $\geq$ 30.0, calculated from self-reported weight and height (weight [kg]/ height [m$^2$]). Pregnant women and respondents reporting weight $<$ 50 pounds or $\geq$ 650 pounds; height $<$ 3 feet or $\geq$ 8 feet; or BMI: $<$12 or $\geq$ 100 were excluded. Alcohol use prevalence was determined from adults who have consumed at least one alcoholic beverage in the past 30 days. All risk factors were based on using the full population (S1–S5 Tables)

Each variable data point at a specific year was linked to the PC rate four years later to account for the least potential lag time for the disease development from the time of exposure [14].

To illustrate trends, incidence rates and prevalence were plotted for each individual state from 1999 to 2016.

### Statistical analysis

Linear mixed-effects models allowing for state specific slopes and intercepts were used to assess the unadjusted time trends over the length of the study period in PC, opioid death rate, obesity, alcohol use and cigarette use. A mixed-effects regression model was then used to examine the multivariate longitudinal association between the change of pancreatic cancer incidence rates and the risk factors included in this study for the years 1999–2016 [15]. This hierarchical model provides a way to estimate covariate effects while controlling for variability in the form of state-specific random effects. Time (years) since 1999 was used to measure annual change and an interaction between time and each risk factor tested for whether the risk factor modified the rate of change. All models used two sided tests at $\alpha = 0.05$ to determine significance.

Additionally, another linear mixed effects model was created using the log transformed opioid death rate.

## Results

From 1999–2016, there were 700,300 incident cases of pancreatic cancer in the United States. Concurrently, 351,630 opioid overdose deaths occurred during this same period [16].

There were marked differences in the distribution of pancreatic cancer incidence and opioid death rates among the states. However, both incidence and death rates increased from 1999 to 2016 in most states (Figs 1 and 2, S1 Fig). The PC incidence rate in the average state increased at an unadjusted rate of 0.137 percentage points per year from 1999 to 2016 (Table 1).

Among contributing factors evaluated at a state level, the opioid death rate increased significantly over the study period (Fig 2B, Beta $\beta = .546$, P-value < 0.0001). We also observed an increase in obesity prevalence (S3A Fig, Beta $\beta = .605$, P-value < 0.0001). We observed no increase or decrease in alcohol prevalence (S3B Fig) ($\beta = -0.0003$, P-value 0.994). Cigarette use prevalence decreased significantly over time (S3C Fig) ($\beta = -0.306$, P-value < 0.0001). Values for each risk factor by state with confidence intervals are provided in the S1–S5 Tables. Data plotted for each of these variables at a national level is shown in S2 Fig.

We employed a mixed-effects regression model to provide a multivariate analysis of the longitudinal associations between the potential risk factors (opioid use, obesity, alcohol and smoking) and the change in incidence rates of pancreatic cancer. This statistical model allows us to observe the subtle changes in the magnitude of the effects of opioid death rate and obesity through time as PC rates are modestly increasing when compared to infectious diseases with dramatic epidemiological drifts. This type of analysis uses opioid death rate over time as the predictor and change in PC rate over time as the outcome.

Results of the mixed effect model are presented in Table 2. The risk factor adjusted PC rate increased over time by an average of 0.108 percentage points per year. The cross-sectional rate of PC was significantly related to the 4-year lagged opioid death rate ($\beta = 0.185$, p<0.0001), while this effect got smaller over time, it still retained a statistically significant association with increasing PC rates. ($\beta = -.019$, p<0.0001). Levels of alcohol use were also associated with higher rates of PC ($\beta = 0.025$, p = 0.007). Obesity rate was not associated with rate of PC ($\beta = 0.023$, p = 0.493), nor degree of change of PC rates ($\beta = 0.002$, p = 0.404). However, there was a

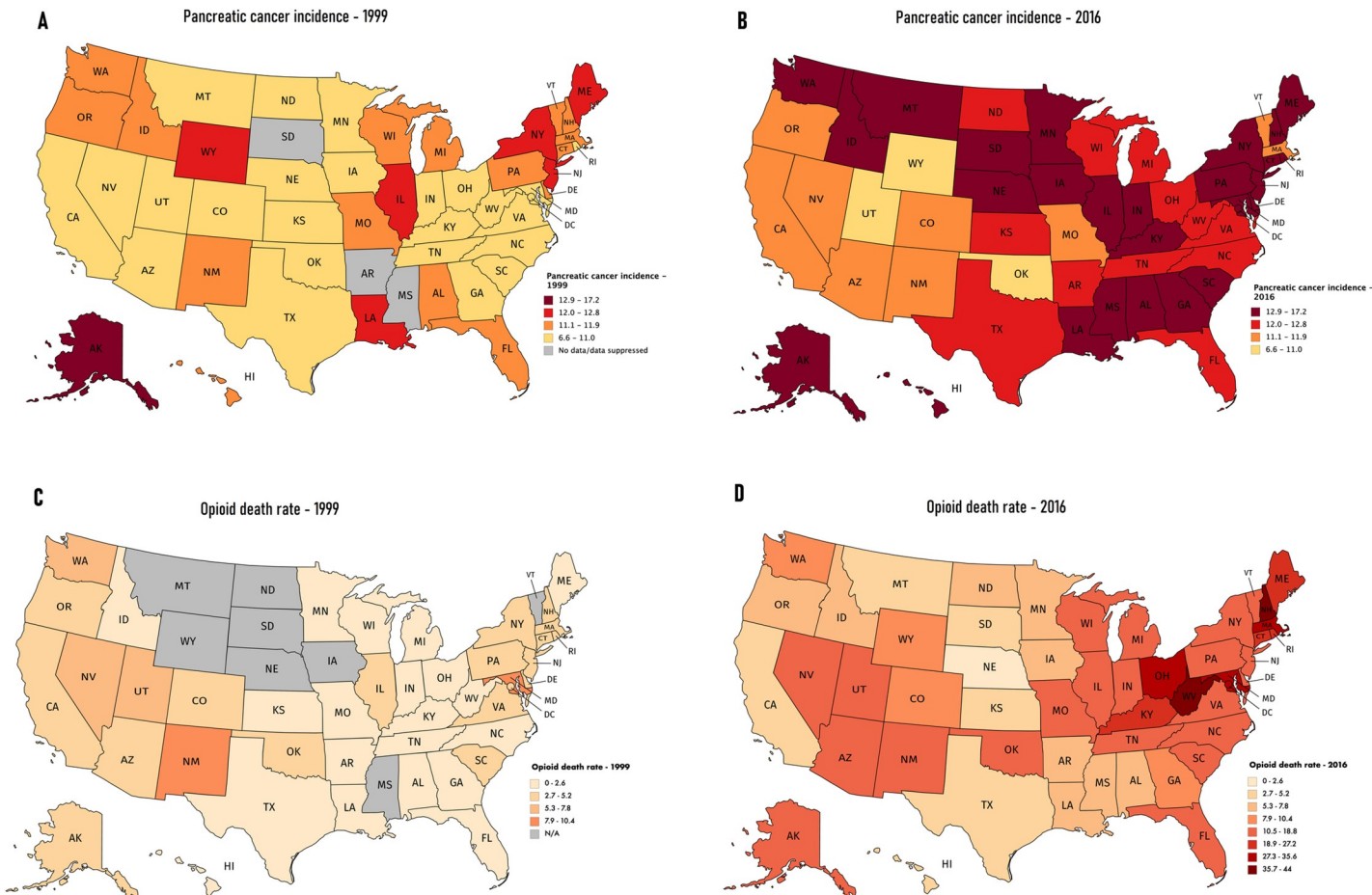

**Fig 1.** Incidence of pancreatic cancer (1999 and 2016), U.S Cancer Statistics Working group, and opioid death rate by state (1999 and 2016), CDC Wonder database (A) Pancreatic cancer incidence visualization by state in 1999, (B) Pancreatic cancer incidence visualization by state in 2016(C) Opioid death rate visualization by state in 1999, (D) Opioid death rate visualization by state in 2016, Reprinted from [MapChart.net] under a CC BY license, with permission from [Minas Giannekas], original copyright [2014].

significant interaction between opioid death rate and obesity prevalence (β = 0.017, p = 0.0002). Indicating rates of PC were higher in states with elevated levels of both obesity and opioid usage 4 years prior. As the opioid death rates overtime was not normally distributed, we also ran our model using the log transformed opioid death rate. As shown in the S6 Table, the output of the model remained unchanged with the log transformed opioid death rate being statistically significantly associated with the pancreatic cancer incidence rate.

To address how changes in opioid death rates were associated with changes in pancreatic cancer rates, we examined the interaction of the opioid death rate with PC rates over time in states with varying level of opioid death rate at the beginning of the study (Fig 2C). Although the overall trend in PC rates is increasing across the states, the rate of the increase is larger for those states starting at lower opioid levels and lower for those that started with higher levels of opioid usage (Fig 2C).

## Discussion

Our results indicate opioid usage, estimated by opioid death rates may partially explain the uptrend of PC rates in the U.S. at the national as well as the state levels over time (Fig 2 and S1

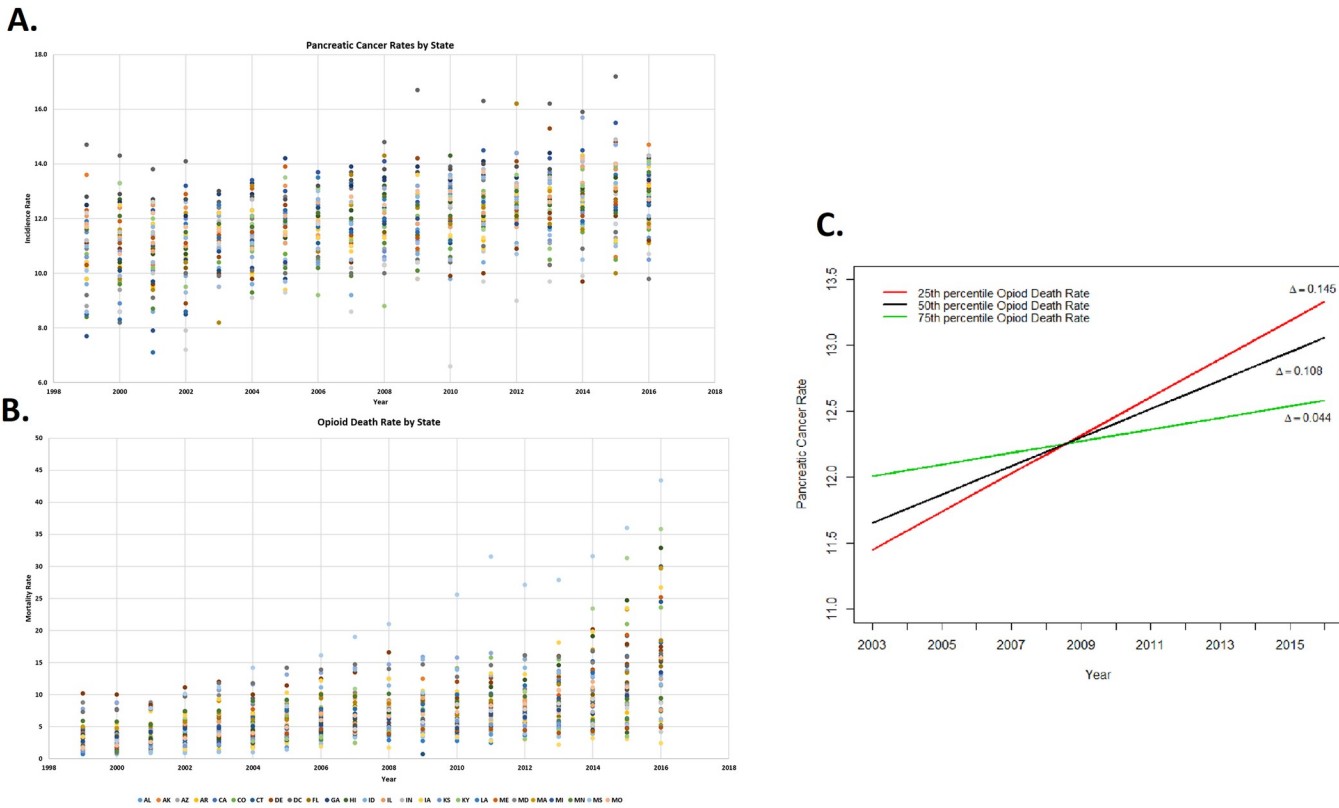

**Fig 2.** (A) Incidence rate of pancreatic cancer over time per 100,000 people, State incidence rate of pancreatic cancer through the years 1999 to 2016, (B) Incidence rate of opioid death overtime per 100,000 people, State incidence rate of opioid death through the years 1999 to 2016, (C) A state with an opioid death rate in the 25th percentile in 1999 had an annual increase in PC that was 34 percent faster than a state at the median opioid death rate and 3.3 times faster than a state with an opioid death rate in the 75th percentile.

Fig). These findings suggest that opioids use may be a novel risk factor for PC, a finding that needs further studies. Once confirmed, implications of the opioids on PC development are clinically significant given the widespread opioid usage in the country. Furthermore, opioids are widely used for pain management in pancreatitis, an established precursor to PC [17, 18] as well as in the cancer itself [19].

We confirmed the increase in the PC rates over time with the majority of the states showing an uptrend in the disease rates over the period of our study (16 years). Opioid use remained a significant predictor to the PC rates although its overall effect became weaker over time, suggesting a ceiling effect. This is supported by our observation that the most increase in PC rates over time occurred in the states that initially had the lowest opioid death rates and showed a significant increase in this opioid index by time (Fig 2). This deeper analysis of lower and

**Table 1. Annual change of PC rate and risk factors.**

| Annual Change of PC and Risk Factors | Estimate | Confidence Interval (95%) |
|---|---|---|
| Pancreatic cancer rate | 0.137 | 0.119, 0.155 |
| Opioid death rate | 0.546 | 0.438, 0.653 |
| Obesity prevalence | 0.605 | 0.571, 0.639 |
| Alcohol prevalence | -0.0003 | -0.075, 0.074 |
| Cigarette use prevalence | -0.306 | -0.333, -0.279 |

**Table 2. Mixed effects regression model analysis results using risk factors with a four year delay to account for the development of pro-cancerous changes.**

| Four-year lagged risk factors | Estimate | Standard Error | P-Value |
|---|---|---|---|
| Opioid death rate | 0.185 | 0.047 | < .0001 |
| Obesity prevalence | 0.023 | 0.033 | 0.493 |
| Alcohol prevalence | 0.025 | 0.009 | 0.007 |
| Cigarette use prevalence | -0.014 | 0.018 | 0.429 |
| Time | 0.108 | 0.021 | < .0001 |
| **Interactions with risk factors with time**[*] | **Estimate** | **Standard Error** | **P-Value** |
| Opioid death rate[*] | -0.019 | 0.004 | < .0001 |
| Obesity prevalence[*] | 0.002 | 0.002 | 0.404 |
| Interaction between opioid death rate and obesity prevalence | 0.017 | 0.004 | 0.0002 |

[*] Denotes that this variable interacted with time in the statistical model.

upper percentile states of opioid death rate and PC rate assists in taking a more nuanced and complex examination of why the effects of opioid death rate on PC rate are decreasing annually despite being a statistically significant predictor.

The association of obesity prevalence and the opioid index in the national databases may be explained by the observation that opioid usage is associated with a poorer lifestyle in general [20]. Such populations could be exposed to other conditions and comorbidities associated with a poor lifestyle that did not enter our model which is a subject of further studies. As suggested by the opposite trend in cigarette use prevalence and PC rates over time in U.S, cigarette use was not associated with PC rate in our analysis.

Opioids and its derivatives are widely used for the management of pain in PC. A large retrospective study found a negative correlation between opioid usage and survival time in patients with unresectable tumors [3]. Population based studies from an area in Iran with high opium consumption rate proposed opium to increase risk of PC [7, 8]. Furthermore a recent post hoc analysis of two randomized controlled trials of patients with advanced cancers (including pancreatic cancer), revealed that those treated frequently with an opioid antagonist had significantly improved overall survival compared to placebo [21].

Mechanistically, opioids as a class of drugs, have been previously studied to have possible carcinogenicity in cancers other than PC [22]. Multiple studies have shown that opioids have a potential to promote cancer progression and metastases in multiple different types of cancers including breast, prostate, lung, esophageal and hepatocellular cancer by various mechanisms including activation of the mTOR pathway, promoting angiogenesis and even promoting epithelial to mesenchymal transition [23]. In addition opioids have been shown to alter the gut microbiome [24] which in turn has been shown to modulate pancreatic carcinogenesis [25]. With these mounting data, the impact of opioids on pancreatic cancer formation and/or progression certainly needs to be further studied.

Using state-based datasets over 16 years, our data suggest opioid usage as a potential risk factor for pancreatic cancer. We used the publicly available data from the CDCs robust reporting system. However, we acknowledge limitations of our analysis. Opioid death rate was used a surrogate for overall opioid use. While prescription opioid use increased in the mid to late 1990s, nonmedical opioid use also increased significantly in the early 2000s and constitutes a large proportion of overall opioid use [26]. Hence, we believe, unlike the opioid prescription rates that miss out the illicit and illegal usage, the opioid death rate index accounts for the pattern of both prescription as well as illicit opioid use across the states. The tested variables showed high variabilities among the fifty states with some missing values per the state-year.

We used a mixed effects model to analyze repeated-measures data over time which has several advantages including accounting for the initial levels of cancer rate as sources of random variability and allows for variation in number of observations across the groups (e.g., the states), allowing all possible data to be used. BRFSS data on obesity (based on height and weight) are self-reported and as such could be subject to misreporting. However, this data could be utilized for estimating trends over time reliably since the same source of self-reported data was consistently used for the trend calculations during the study period here. Despite the limitations of the CDC databases, they are currently the most reliable datasets at our disposal for analyzing the interactions between opioid use and pancreatic cancer on a national scale. We used the values with 95% confidence intervals from these datasets; while we are not able to independently ascertain these values, using multiple data points overtime, we were able to find the interplay between the increasing pancreatic cancer rates in relation to the increase in opioid use.

We acknowledge that our study is subject to ecological limitations; association between opioid usage and pancreatic cancer rates in our study is observed at group (states and national) levels and may not be directly extrapolatable to the individual level. In the absence of longitudinal dataset that reliably register long-term outcomes in opioid users, our observation needs to be further tested in future cohorts where individual-level data on opioid consumption and long-term outcome is available. However, we corrected in our analysis for any other known confounding factors that could contribute to the risk of pancreatic cancer in our examined populations.

In summary, our analysis of state-based databases suggests that opioid usage pattern may explain the trend of increased pancreatic cancer over time. Opioids as a novel risk factor pancreatic cancer needs to be confirmed in population-based studies.

## Supporting information

**S1 Table. Pancreatic cancer incidence rate by state, United States cancer statistics: Data visualizations.**
(DOCX)

**S2 Table. Opioid death rate by state, United States cancer statistics: Data visualizations.**
(DOCX)

**S3 Table. Obesity prevalence by state (%), Behavioral Risk Factor Surveillance System (BRFSS).**
(DOCX)

**S4 Table. Alcohol use prevalence by state (%), Behavioral Risk Factor Surveillance System (BRFSS).**
(DOCX)

**S5 Table. Cigarette use by state (%), Behavioral Risk Factor Surveillance System (BRFSS).**
(DOCX)

**S6 Table. Mixed effects regression model using log transformed opioid death rate.**
(DOCX)

**S1 Fig.** (A) Incidence rate of pancreatic cancer over time per 100,000 people, National incidence rate of pancreatic cancer through the years 1999 to 2016, (B) Incidence rate of opioid death overtime per 100,000 people, National incidence rate of opioid death rate through the years 1999 to 2016.
(TIF)

**S2 Fig.** (A) National prevalence of obesity through the years 1999 to 2018, (B) National prevalence (%) of alcohol use through the years 1999 to 2018, (C) National prevalence (%) of tobacco use through the years 1999 to 2017.
(TIF)

**S3 Fig.** (A) State prevalence of obesity overtime, Prevalence of obesity by state through the years 1999 to 2018, (B) Prevalence of alcohol use by state through the years 1999 to 2018, (C) Prevalence of tobacco use by state through the years 1999 to 2017.
(TIF)

## Author Contributions

**Conceptualization:** Usman Barlass, Faraz Bishehsari.

**Data curation:** Ameya Deshmukh.

**Formal analysis:** Usman Barlass, Ameya Deshmukh, Todd Beck, Faraz Bishehsari.

**Funding acquisition:** Faraz Bishehsari.

**Investigation:** Usman Barlass, Faraz Bishehsari.

**Methodology:** Usman Barlass, Todd Beck, Faraz Bishehsari.

**Supervision:** Faraz Bishehsari.

**Writing – original draft:** Usman Barlass, Ameya Deshmukh, Todd Beck, Faraz Bishehsari.

**Writing – review & editing:** Usman Barlass, Ameya Deshmukh, Todd Beck, Faraz Bishehsari.

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
