## [Decision Letter · Decision Letter 0]

22 Sep 2020

PONE-D-20-20071

Opioid use as a potential contributing factor to the rise of pancreatic cancer; an analysis of state and national level databases from the United States

PLOS ONE

Dear Dr. Bishehsari,

Thank you for submitting your manuscript to PLOS ONE. After careful consideration, we feel that it has merit but does not fully meet PLOS ONE’s publication criteria as it currently stands. Therefore, we invite you to submit a revised version of the manuscript that addresses the points raised during the review process.

We look forward to receiving your revised manuscript.

Kind regards,

Ram K. Raghavan

Academic Editor

PLOS ONE

Journal Requirements:

2. In your ethics statement in the Methods section and in the online submission form, please confirm that all data were fully anonymized before you accessed them.

3. Please include the date(s) on which you accessed the database(s) or record(s) to obtain the data used in your study.

4.PLOS requires an ORCID iD for the corresponding author in Editorial Manager on papers submitted after December 6th, 2016. Please ensure that you have an ORCID iD and that it is validated in Editorial Manager. To do this, go to ‘Update my Information’ (in the upper left-hand corner of the main menu), and click on the Fetch/Validate link next to the ORCID field. This will take you to the ORCID site and allow you to create a new iD or authenticate a pre-existing iD in Editorial Manager. Please see the following video for instructions on linking an ORCID iD to your Editorial Manager account: https://www.youtube.com/watch?v=_xcclfuvtxQ

5. Please include your tables as part of your main manuscript and remove the individual files. Please note that supplementary tables (should remain/ be uploaded) as separate "supporting information" files

6.We note that [Figure(s) 1] in your submission contain [map/satellite] images which may be copyrighted. All PLOS content is published under the Creative Commons Attribution License (CC BY 4.0), which means that the manuscript, images, and Supporting Information files will be freely available online, and any third party is permitted to access, download, copy, distribute, and use these materials in any way, even commercially, with proper attribution. For these reasons, we cannot publish previously copyrighted maps or satellite images created using proprietary data, such as Google software (Google Maps, Street View, and Earth). For more information, see our copyright guidelines: http://journals.plos.org/plosone/s/licenses-and-copyright.

1.    You may seek permission from the original copyright holder of Figure(s) [1] to publish the content specifically under the CC BY 4.0 license. 

Reviewers' comments:

Reviewer's Responses to Questions

**Comments to the Author**

1. Is the manuscript technically sound, and do the data support the conclusions?

Reviewer #1: Partly

Reviewer #2: Yes

2. Has the statistical analysis been performed appropriately and rigorously? 

Reviewer #1: Yes

Reviewer #2: Yes

3. Have the authors made all data underlying the findings in their manuscript fully available?

Reviewer #1: Yes

Reviewer #2: Yes

4. Is the manuscript presented in an intelligible fashion and written in standard English?

Reviewer #1: Yes

Reviewer #2: Yes

5. Review Comments to the Author

Reviewer #1: The authors present results of a study using state and national level databases to assess the association between opioid use and pancreatic cancer. Using mixed effects models and state level data across 16 years of observation, authors find that opioid death rates are associated with incident pancreatic cancer rates 4 years later as well as change over time in the incident pancreatic cancer rates. They conclude that opioid use may account for some of the increase in pancreatic cancer. The manuscript will be strengthened if the authors consider the following points.

1. Authors are using estimated values from a variety of sources (and include 95% confidence intervals in the supplemental tables). Did the authors use the uncertainty in these values in any way in their analyses? At least some mention of this as a limitation would be good.

2. Were the assumptions of normality met for the outcomes evaluated with the mixed effects regression model? I am wondering about the opioid death rates in particular.

3. In the results, authors interpret the multivariate model results and in particular mention that obesity rate was not associated with the rate of pancreatic cancer. However, there is an interaction in the model (between obesity rate and opioid death rate), which makes the interpretation of the main effect not as straight forward. Authors should clarify their interpretation. Also, it is not clear from the table whether this interaction was an interaction with time. The coefficient in the text differs slightly than that in the table, so that should be corrected. The sentence in the text reporting that interaction and the sentence following it likely should be combined (as the last sentence is an incomplete sentence).

4. The authors used lagged risk factors to predict the rates of pancreatic cancer. Did authors consider assessing how change in opioid death rates were associated with change in pancreatic cancer rates?

5. Do the authors have any comments about the limitations of ecologic analyses in the context of their question?

Minor points:

1. In the abstract, the authors give the result of how the opioid death rate predicted incident pancreatic cancer. Authors might want to clarify here that they are predicting incident pancreatic cancer 4 years later. Also, in the next line, authors talk about the significant effect on the estimated annual change. The earlier sentence refers to the initial level of a state's opioid death rate, but they appear to use the time-varying 4-year lagged variable, so the effect on change over time is not restricted to just the initial level.

2. Under Data collection: "This data was" should be "These data were" and authors refer to Tables 1-5, but I think they mean Supplemental Tables 1-5. ("this data" also appears in the Discussion section and should "these data").

3. Figure 1: Panels C and D use the same colors, but they have different meanings. Since authors likely want readers to visually compare the two graphs to see the changes in opioid death rates, authors should consider having the same rate categories and colors (where applicable) and then have additional categories/colors to more easily see the differences.

Reviewer #2: the highlighted text should be edited and rewrite. also potential ecologic fallacy is one of the limitations of the current study that should be addressed in the discussion. figures in the text had not good resolution.

6. PLOS authors have the option to publish the peer review history of their article (what does this mean?). If published, this will include your full peer review and any attached files.

Reviewer #1: No

Reviewer #2: No

---

## [Author Response · Author response to Decision Letter 0]

29 Oct 2020

We would like to thank the editor and reviewers for taking time to review our manuscript. We have responded to all feedback in our "Response to comments" document uploaded with the re-submission. Thank you.

---

## [Editor Report · Decision Letter 1]

8 Dec 2020

Opioid use as a potential risk factor for pancreatic cancer in the United States; an analysis of state and national level databases

PONE-D-20-20071R1

Dear Dr. Bishehsari,

We’re pleased to inform you that your manuscript has been judged scientifically suitable for publication and will be formally accepted for publication once it meets all outstanding technical requirements.

Kind regards,

Ram K. Raghavan

Academic Editor

PLOS ONE
---

## [Editor Report · Acceptance letter]

11 Dec 2020

PONE-D-20-20071R1 

Opioid use as a potential risk factor for pancreatic cancer in the United States; an analysis of state and national level databases 

Dear Dr. Bishehsari:

I'm pleased to inform you that your manuscript has been deemed suitable for publication in PLOS ONE. Congratulations! Your manuscript is now with our production department. 

Kind regards, 

on behalf of

Dr. Ram K. Raghavan 

Academic Editor

PLOS ONE